# Mediator complex subunit MED23 dampens antiviral innate immunity by restricting RIG-I expression

**De-fei Xiong, Yi-yang Zhang, Zhi-chao Wang, Yuan-ming Zheng, Han-qing Zhang, Gang Wang** *

State Key Laboratory of Genetic and Development of Complex Phenotypes, School of Life Sciences and Zhongshan Hospital, Fudan University, Shanghai, China

* gwang_fd@fudan.edu.cn

## Abstract

The detection of cytosolic viral nucleic acids via pattern-recognition receptors (PRRs) activates multiple signaling pathways, leading to the production of interferons (IFNs), which are essential for host survival during viral infection. Precise control of PRR gene expression is crucial for maintaining immune homeostasis. Here, we showed that Mediator complex subunit 23 (Med23) is required for the precise production of the innate immune receptor RIG-I in response to RNA virus infection. Med23 deficiency markedly enhances the production of IFN-I, proinflammatory cytokines, and IFN-stimulated genes (ISGs) in both multiple cell lines (MEFs, RAW264.7 cells, and HeLa cells) and mouse primary macrophages (bone marrow-derived macrophages [BMDMs] and peritoneal macrophages [PEMs]) infected with RNA virus VSV or stimulated with poly(I:C). Myeloid-specific Med23 knockout mice were generated to test the critical role of Med23 in host resistance to VSV infection in vivo. Mechanistically, Med23 interacts with the transcription factor forkhead box O3 (Foxo3) to negatively regulate RIG-I, thereby modulating IFN-I signaling. Collectively, these findings elucidate a previously unrecognized role of Med23 as a gatekeeper of the RIG-I-mediated antiviral innate immune response and suggest a potential target for controlling viral infection.

## Introduction

Infections caused by RNA viruses pose a significant threat to public health. The recent coronavirus disease 2019 (COVID-19) is caused by severe acute respiratory syndrome coronavirus 2 (SARS-CoV-2), which is transmitted among various animals and is spreading globally [1,2]. However, the pathogenic mechanisms underlying rapid viral infection are not yet fully understood, and the related host defense mechanisms remain to be actively investigated.

**Data availability statement:** All relevant data are within the paper and its Supporting information files. All RNA-seq files are available from the GEO database (accession number GSE281340).

**Funding:** This work was supported in part by grants from the National Natural Science Foundation Grant of China (32030028 and 32170735) to GW and the Ministry of Science and Technology of China (2022YFA0806200) to GW. The funders had no role in study design, data collection and analysis, decision to publish, or preparation of the manuscript.

**Competing interests:** The authors have declared that no competing interests exist.

**Abbreviations :** Act D, Actinomycin D; BMDMs, bone marrow-derived macrophages; CARD, caspase-recruiting domain; DCs, dendritic cells; GEO, Gene Expression Omnibus; GFP, green fluorescent protein; GO, Gene Ontology; H&E, hematoxylin and eosin; IFNs, interferons; ISGs, IFN-stimulated genes; ISREs, interferon-stimulated response elements; MAVS, mitochondrial antiviral signalling; Med23, Mediator complex subunit 23; MEFs, mouse embryonic fibroblasts; NK, natural killer; PEMs, peritoneal macrophages; Pol II, polymerase II; PRRs, pattern-recognition receptors; qRT-PCR, quantitative reverse transcription PCR; RIG-I, retinoic acid-inducible gene I; RNA-seq, RNA sequencing; TBK1, TANK binding kinase 1; VSV, vesicular stomatitis virus.

The innate immune system provides the first line of defense to protect the host from pathogens [3]. Upon viral infection, innate immunity is activated by the recognition of pathogen-associated molecular patterns derived from viral products during infection through host cellular pattern-recognition receptors (PRRs) [4]. Several PRRs, notably retinoic acid-inducible gene I (RIG-I) and melanoma differentiation-associated gene-5, are responsible for detecting pathogen-derived RNA in the cytoplasm [5,6]. RIG-I contains a tandem caspase-recruiting domain (CARD) that initiates downstream signaling and a central DExD/H box RNA helicase domain that recognizes viral RNA [7,8]. When pathogenic RNA binds to the helicase domain, RIG-I is recruited to the mitochondrial antiviral signaling (MAVS) protein via the CARD domain [9]. MAVS then activates downstream adaptors, including TANK binding kinase 1 (TBK1), IκB kinase ε (IKKε), interferon regulatory factor 3 (IRF3), and nuclear factor κB (NF-κB) [5]. This activation results in the production of IFN-I and proinflammatory cytokines. These IFNs trigger the JAK/STAT signaling pathway to increase the expression of many multifunctional IFN-stimulated genes (ISGs) and eventually establish an antiviral state that provides innate immunity for host cells [10]. Precise regulation of RIG-I signaling is crucial for effective viral clearance without causing immunopathology. Studies have increasingly focused on regulating RIG-I functions through posttranslational modifications such as phosphorylation [11,12], SUMOylation [13], and polyubiquitination [14–16]. However, the involvement of nuclear molecules in antiviral innate immune processes has received less attention [17].

The Mediator complex, a master cofactor in the cell nucleus, links transcription factors and RNA polymerase II (Pol II) in transcription regulation [18], particularly at enhancer-promoter sites [19]. Different Mediator subunits can activate or suppress gene transcription by interacting with specific transcription factors in response to environmental or developmental cues [20]. Evidence suggests that Mediator complex subunit 23 (Med23) regulates diverse biological processes, including developmental abnormalities [21], metabolic disorders [22], and cancers [23]. Our previous study revealed that hepatic Med23 is a key modulator of chemokine production and inflammatory responses, potentially serving as a novel target for clinical intervention in liver fibrosis [24]. Additionally, Med23 plays a crucial role in controlling T-cell activation and enhancing antitumor T-cell function [25]. Although Med23 is known to be vital for the immune response, its exact functionality in antiviral innate immunity and viral defense remains unexplored.

In this study, we initially reported that silencing Med23 in multiple cell lines and primary macrophages inhibits RNA virus replication by promoting the expression of IFN-I and ISGs, suggesting that Med23 plays an important role in antiviral immunity. We generated myeloid-specific Med23 knockout mice and demonstrated that Med23 is essential for host resistance to RNA virus infections in vivo. We further revealed a molecular mechanism through which Med23 negatively regulates the expression of the innate immune receptor RIG-I by cooperating with the transcription factor Foxo3, which has been shown to be involved in limiting the inflammatory sequelae of

antiviral responses [26,27]. These findings revealed an unexpected role for Med23 in host antiviral immune responses and highlighted the role of transcriptional cofactors in controlling viral infections.

## Results

### Med23 deficiency enhances type I interferons and cytokine production in response to viral infection

To investigate the function of Med23 in antiviral innate immune responses, we first infected Med23 knockout mouse embryonic fibroblasts (MEFs) with the RNA virus vesicular stomatitis virus (VSV) and measured the production of IFN-I and proinflammatory cytokines (S1A Fig). Interestingly, Med23 KO MEFs produced high mRNA levels of IFNβ, IFNα2, IL6, and CCL5 following VSV infection for the indicated times (Fig 1A). Poly(I:C) is a synthetic double-stranded RNA analog used in preclinical models to simulate viral RNA replication intermediates. The mRNA levels of these cytokines were greater in Med23 knockout MEFs than in WT MEFs after poly(I:C) stimulation (Fig 1B). The expression of IFNβ proteins induced by VSV or poly(I:C) were significantly higher in Med23 KO MEFs than WT(Fig 1C and 1D). However, IFNβ, IFNα2, IL6, and CCL5 mRNA levels, as well as the excretion of IFNβ stimulated by VSV or poly(I:C) are partially repressed by re-expressing exogenous Med23 in Med23 knockout MEFs (Figs 1E–1G and S1B–S1C). To confirm the impact of Med23 on IFN-I and proinflammatory cytokine production, we generated Med23-ablated RAW264.7 macrophage lines and HeLa cell lines via the CRISPR/Cas system. We observed increased mRNA expression of these cytokines in VSV-challenged Med23 KO RAW264.7 cells compared with WT cells (S1D–S1E Fig). Compared with WT HeLa cells, Med23-KO HeLa cells also presented increased type I interferon and proinflammatory cytokine mRNA expression in response to either VSV infection or poly(I:C) stimulation (S1F–S1H Fig).

To further investigate the impact of Med23 on the virus-induced IFN-I response, we used IFN-β luciferase reporters to evaluate the regulatory effect of Med23 on IFN-I signaling through VSV infection or poly(I:C) stimulation. As shown in Figs 1H and 1I, IFN-β luciferase activity was significantly induced in Med23-deficient HeLa cells. Furthermore, re-expression of the Med23 protein in Med23 KO cells markedly inhibited VSV- or poly(I:C)-induced IFN-β promoter-driven luciferase activity in a dose-dependent manner (S1I Fig). These results demonstrate that Med23 plays a crucial role in regulating the IFN-I response following viral infection.

We next explored the impact of Med23 on the VSV-induced innate immune pathway. We detected a significant increase in the phosphorylation of TBK1 and IRF3 in Med23 knockout MEFs stimulated with VSV or poly(I:C) (Fig 1J and 1K). Similar results were observed in Med23 KO Raw264.7 or HeLa cells (S1J–S1L Fig). After activation, phosphorylated IRF3 translocates from the cytoplasm to the nucleus to activate the transcription of IFN-I and proinflammatory cytokines. We examined whether the nuclear translocation of IRF3 in response to VSV infection or poly(I:C) stimulation was affected by Med23 deficiency. Consistent with the increased phosphorylation of IRF3, the level of IRF3 nuclear translocation was significantly greater in Med23 knockout cells than in WT cells after VSV infection or poly(I:C) stimulation (Figs 1L and S1M). Together, these data suggest that Med23 controls the immune escape of VSV by regulating the activation of IFN-I signaling.

To further elucidate the function of Med23 in defense against RNA virus infection, we infected WT or Med23 knockout cells with VSV-expressing green fluorescent protein (GFP) and monitored VSV replication via flow cytometry analysis of the GFP-positive cells. Flow cytometry revealed that VSV replication was substantially inhibited in Med23 KO MEFs, suggesting that Med23 deficiency suppressed RNA virus replication in vitro (Fig 1M). Meanwhile, the titers of VSV were decreased in Med23 KO MEFs (S1N Fig). Re-expression of the Med23 protein in Med23 knockout cells reversed RNA virus replication (Figs 1N and S1O). Consistent with these findings, Med23 deficiency inhibited RNA virus replication in RAW264.7 cells and HeLa cells (S1P and S1Q Fig). Collectively, these findings indicate that Med23 deficiency leads to increased production of type I interferons and innate cytokines in response to viral infection, thereby inhibiting viral replication.

 

PLOS Biology

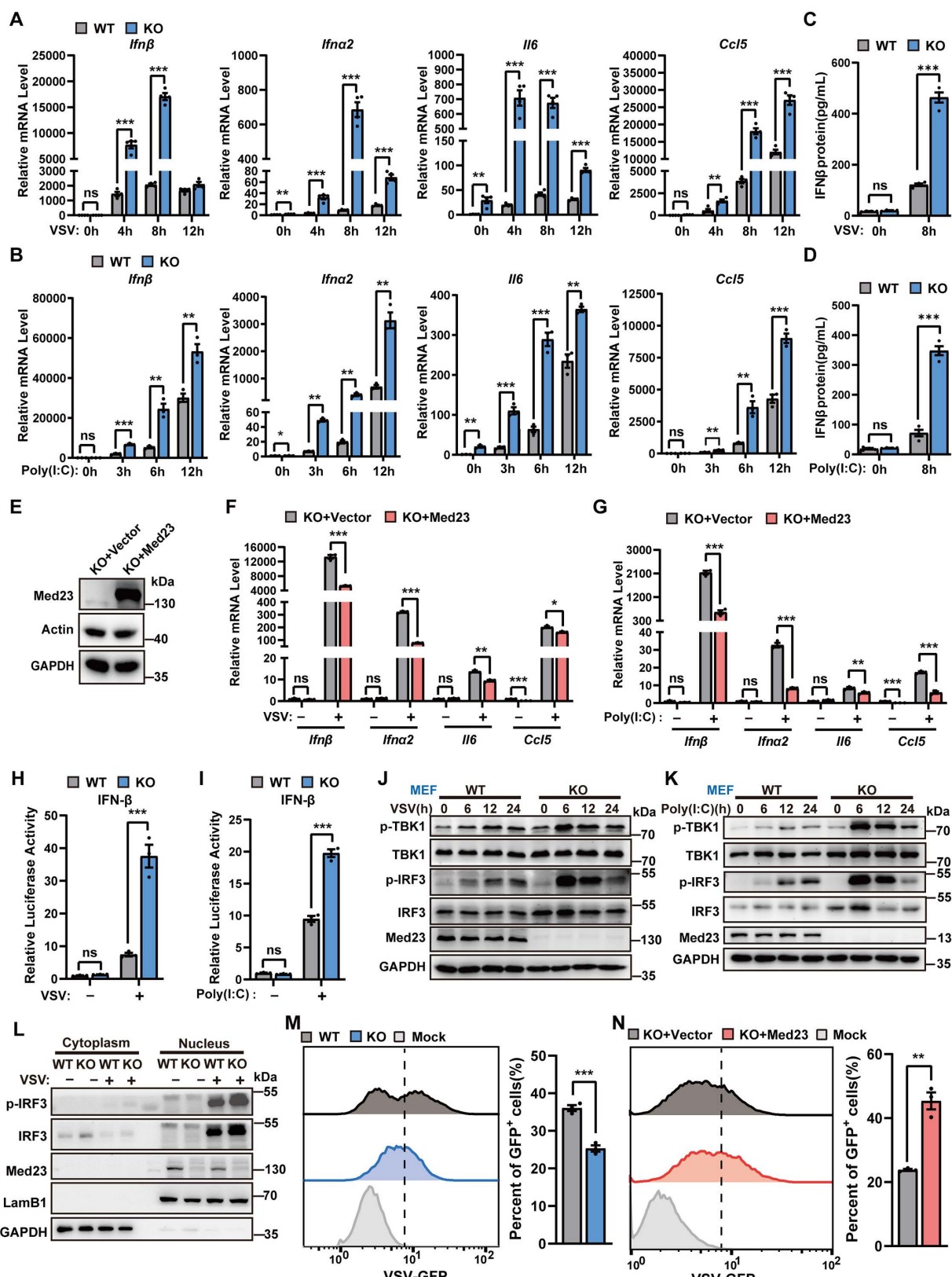

**Fig 1. Increased expression of type I interferons and proinflammatory cytokines after the knockout of Med23. (A)** and **(B)** qPCR analysis of the levels of the indicated genes in Med23-knockout MEFs infected with VSV (MOI = 1, $n = 4$) or stimulated with poly(I:C) (5 μg/mL, $n = 3$) indicated times.

Expression levels were normalized to GAPDH mRNA and then to the 0 h WT sample. **(C)** and **(D)** ELISA analysis of IFNβ in the supernatant of WT and Med23 KO MEFs (n = 4) infected with VSV (MOI = 1) or stimulated with poly(I:C) (5 μg/mL) for 8 h. **(E)** Immunoblotting to detect Med23 in Med23 knockout MEFs reconstituted with pMSCV-Med23 retroviral overexpression system. **(F)** and **(G)** qPCR analysis of the levels of the indicated genes in Med23-overexpressing KO MEFs (n = 3) infected with VSV (MOI = 1) or stimulated with poly(I:C) (5 μg/mL) for 8 h. Expression levels were normalized to GAPDH mRNA expression and then to the 0 h WT sample. **(H)** and **(I)** Luciferase reporter assays were conducted with WT and Med23-knockout HeLa cells (n = 3) transfected for 48 h with IFN-β reporter plasmids and then infected with VSV (MOI = 1) or stimulated with poly(I:C) (5 μg/mL) for 8 h. The results are presented relative to the luciferase activity in WT cells without VSV infection or poly(I:C) stimulation. **(J)** and **(K)** Immunoblot analysis of the indicated protein in Med23-knockout MEFs infected with VSV (MOI = 1) or stimulated with poly(I:C) (5 μg/mL) for 8 h. **(L)** Immunoblot analysis of p-IRF3 protein in the nuclear and cytoplasmic fractions of WT and Med23-knockout HeLa cells upon infection with VSV(MOI = 1) for 8 h. GAPDH served as a cytoplasmic control. Lamin B1 served as a nuclear protein control. **(M)** Flow cytometry analysis of GFP fluorescence intensity in WT and Med23-knockout MEFs (n = 3) infected with VSV-GFP (MOI = 1) for 8 h. **(N)** Flow cytometry was used to detect the GFP fluorescence intensity of Med23-overexpressing knockout MEFs (n = 3) infected with VSV-GFP (MOI = 1) for 8 h. Data are presented as means ± SEM of at least three independent experiments. Statistical analysis in(A–D),(F–I), and (M–N) was performed by two-tailed unpaired Student t test; ns, not significant, *P < 0.05, **P < 0.01, and ***P < 0.001. The data underlying this figure can be found in S1 Data.

## Med23-deficient macrophages exhibit an enhanced antiviral innate response

Considering the specific role of Med23 in the immune response and the importance of macrophages as key innate immune cells, we generated myeloid-specific Med23 conditional knockout mice. Med23-floxed mice were crossed with transgenic mice that express Cre recombinase under the control of the LysM promoter to generate Med23^f/f LysM-Cre mice (referred to as Med23^ΔM mice) (S2A Fig). The successful deletion of Med23 was confirmed by RT-qPCR and immunoblot analysis of BMDMs or PEMs isolated from Med23^ΔM mice (Figs 2A and S2B). Flow cytometry was performed to detect the proportions of BMDMs, PEMs, and B cells, CD4+ T cells, CD8+ T cells, natural killer (NK) cells, neutrophils, dendritic cells (DCs), macrophages in the spleens of Med23^f/f and Med23^ΔM mice. The results revealed no changes in the populations of immune cell subsets in Med23^ΔM mice (S2C–S2J Fig).

We then quantified the VSV virus-induced production of IFN-I and proinflammatory cytokines in murine primary macrophages derived from Med23^f/f or Med23^ΔM mice. Consistently, the deletion of Med23 in macrophages significantly increased the mRNA levels of IFN-I, IL-6, and CCL5 in both BMDMs and PEMs (Fig 2B and 2C). Compared with their Med23^f/f counterparts, Med23-deficient BMDMs and PEMs presented increased phosphorylation of TBK1 and IRF3 (Fig 2D and 2E), which was associated with a reduced VSV mRNA level and virus titers (Fig 2F and 2G). These data demonstrate that Med23 negatively regulates RNA virus-induced IFN-I signaling in primary macrophages.

## Mice with macrophage-specific Med23 deficiency are protected from RNA virus infection

To elucidate the crucial role of Med23 in innate immunity in vivo, we intraperitoneally infected Med23^ΔM mice and their littermate control Med23^f/f mice with VSV and quantified the expression of IFN-β and IL6 mRNAs in the spleen, lungs, and liver. Consistent with the in vitro findings, qPCR analysis revealed elevated levels of IFN-β and IL6 mRNA in the spleen, lungs of Med23^ΔM mice than in those of Med23^f/f mice, with a similar trend in the liver (Fig 3A–3C). Furthermore, the VSV-G mRNA expression and virus titers was significantly lower in Med23^ΔM mice than in Med23^f/f mice (Fig 3D), suggesting that Med23 plays a significant role in host antiviral innate responses in vivo.

VSV infection can cause lung lesions and damage in mice, with noticeable thickening of the alveolar walls [28]. We hypothesize that Med23^ΔM may reduce lung damage caused by VSV infection, as Med23 deficiency in macrophages could enhance innate immunity. In the absence of viral infection, Med23 knockout did not affect the lung tissue of the mice (Fig 3E). In contrast, compared with control mice, Med23^ΔM mice presented a reduced incidence of VSV-induced pathological lesions in the lung, and lower lung injure score (Fig 3E). Furthermore, Med23^ΔM mice exhibited less body weight loss upon low-dose VSV infection (Fig 3F), and had significantly less mortality than Med23^f/f mice upon high-dose infection (Fig 3G).

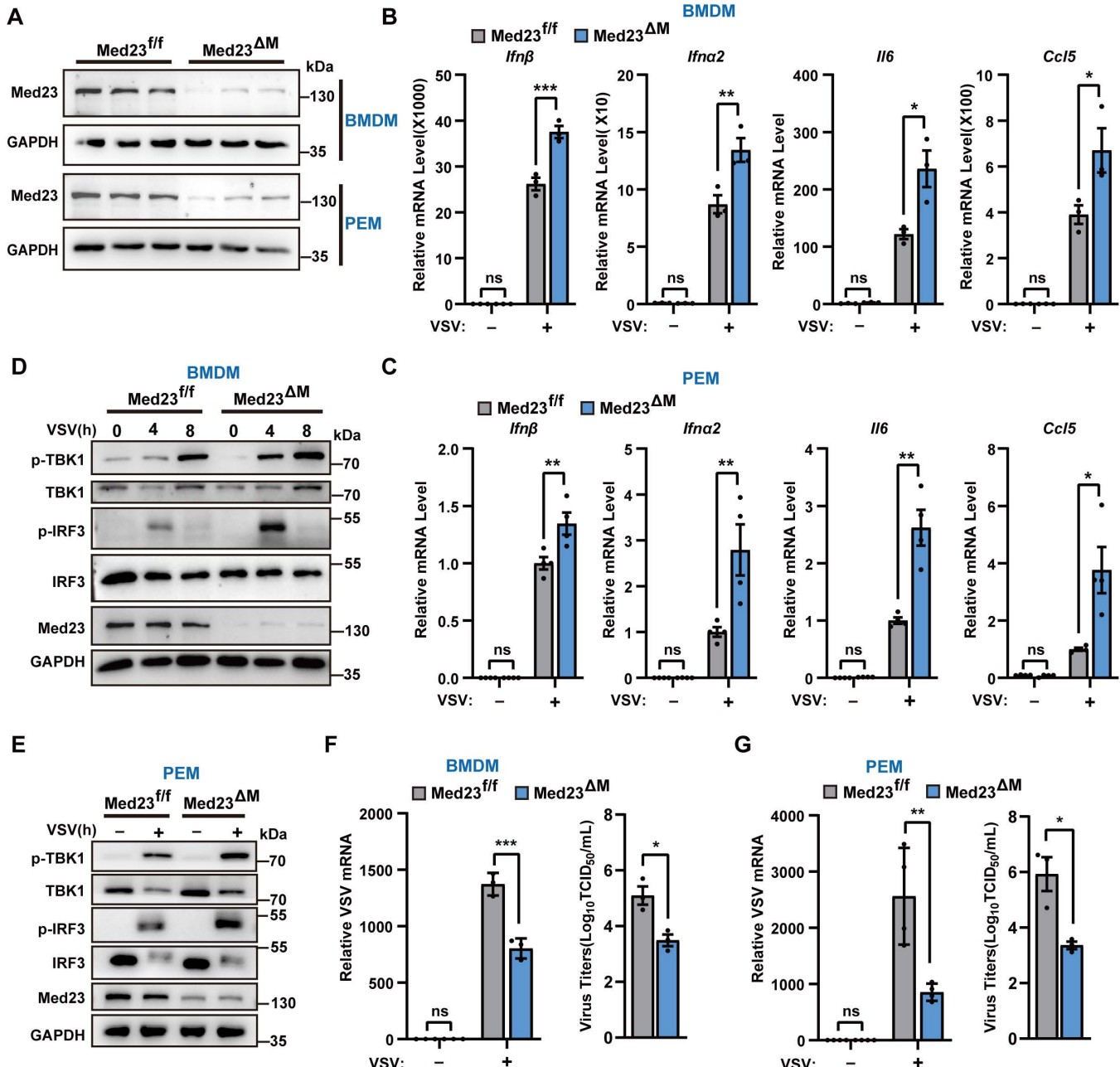

**Fig 2. Increased production of type I interferons and proinflammatory cytokines in Med23-deficient macrophages. (A)** Western blotting analysis of Med23 protein expression in BMDMs and PEMs from Med23$^{f/f}$ and Med23$^{\Delta M}$ mice ($n=3$). **(B)** and **(C)** qPCR analysis of the indicated genes in Med23$^{f/f}$ and Med23$^{\Delta M}$ BMDMs or PEMs ($n=3$) infected with VSV(MOI=0.5, $n=3$) for 4 h. **(B)** Expression levels were normalized to GAPDH mRNA expression and then to the Med23$^{f/f}$ BMDMs without VSV infection. **(C)** Expression levels were normalized to GAPDH mRNA expression and then to the Med-23$^{f/f}$ PEMs with VSV infection. **(D)** and **(E)** Immunoblot analysis of the indicated proteins in Med23$^{f/f}$ and Med23$^{\Delta M}$ BMDMs or PEMs infected with VSV (MOI=0.5). **(F)** and **(G)** Expression of VSV-G mRNA and TCID$_{50}$ assay of virus titers after infection of Med23$^{f/f}$ and Med23$^{\Delta M}$ BMDMs or PEMs (MOI=0.5, $n=3$). VSV-G mRNA levels were normalized to GAPDH mRNA expression and then to the Med23$^{f/f}$ BMDMs or PEMs without VSV infection. Data present means±SEM of three independent experiments. Statistical analysis in (B and C), and (F and G) was performed by two-tailed unpaired Student's $t$ test; ns, not significant, *$P<0.05$, **$P<0.01$, and ***$P<0.001$. The data underlying this figure can be found in S1 Data.

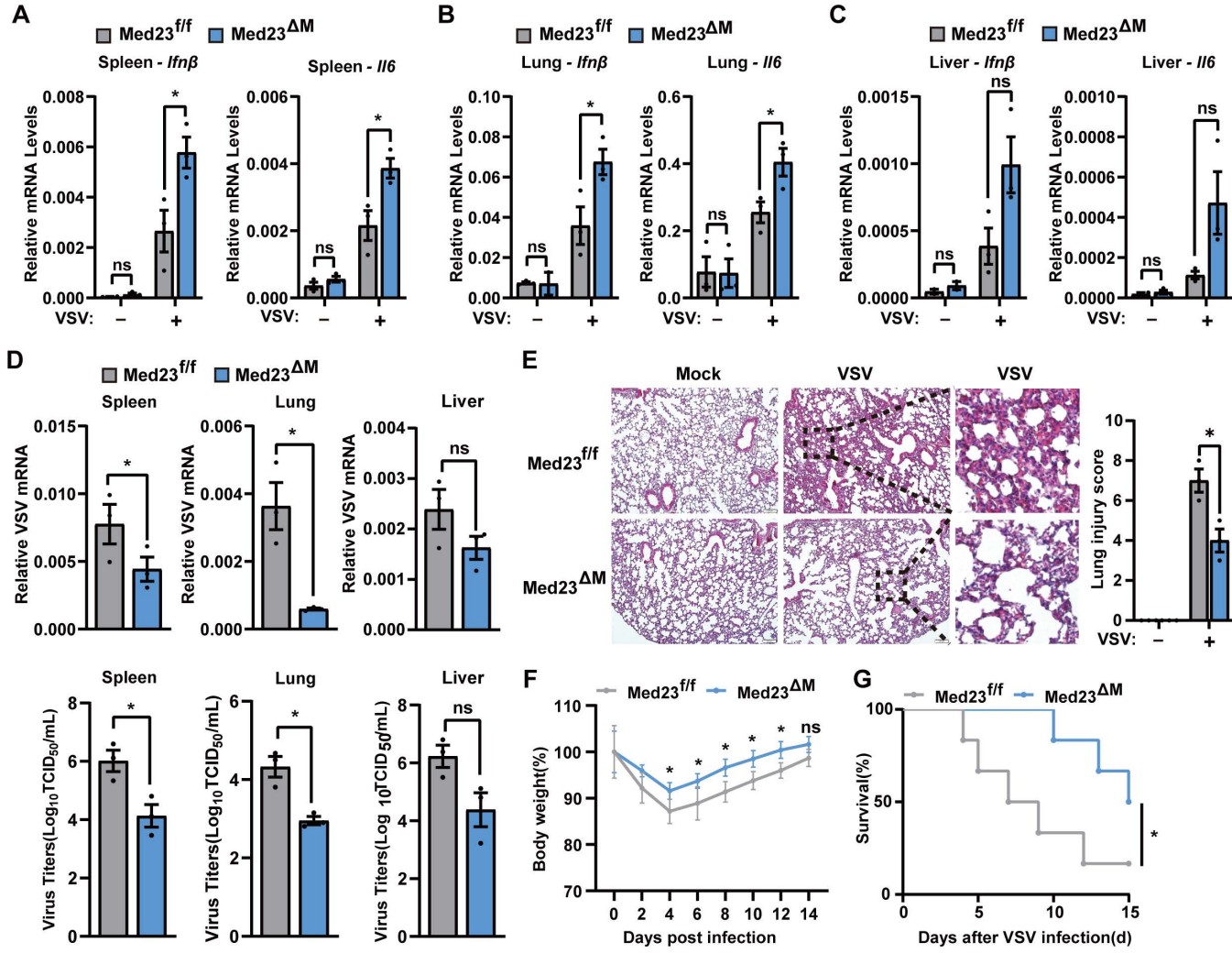

**Fig 3. Med23 deficiency attenuates viral replication in vivo after virus infection. (A–C)** Med23$^{f/f}$ and Med23$^{\Delta M}$ mice were infected for 24 h via the intraperitoneal injection of VSV ($2 \times 10^7$ PFU per mouse, $n = 3$), and then, qPCR analysis of the Ifnβ and Il6 levels in the spleen, liver, and lungs was performed. Expression levels were normalized to GAPDH mRNA expression. **(D)** qPCR analysis of the number of VSV copies and TCID$_{50}$ analysis of virus titers in the spleen, liver, and lungs of Med23$^{f/f}$ and Med23$^{\Delta M}$ mice 24 h after infection ($2 \times 10^7$ PFU per mouse, $n = 3$). VSV-G mRNA levels were normalized to GAPDH mRNA expression. **(E)** Hematoxylin and eosin-stained sections of lungs from Med23$^{f/f}$ and Med23$^{\Delta M}$ mice uninfected or infected with VSV ($2 \times 10^7$ PFU per mouse, $n = 3$) for 24 h, and the lung injury score was calculated. Scale bars: 100μm. **(F)** body weight was monitored in Med-23$^{f/f}$ and Med23$^{\Delta M}$ mice injected intraperitoneally with VSV ($2 \times 10^7$ PFU per mouse, $n = 4$) for 14 days. **(G)** Survival (Kaplan-Meier curves) was monitored in Med23$^{f/f}$ and Med23$^{\Delta M}$ mice injected intraperitoneally with VSV ($1 \times 10^8$ PFU per mouse, $n = 6$) for 15 days. Data present means ± SEM of at least three independent experiments. Statistical analysis was performed by two-tailed unpaired Student's $t$ test; ns, not significant, $*P < 0.05$. The data underlying this figure can be found in S1 Data.

Collectively, these results suggest that Med23-deficient mice showed an enhanced innate immune response to RNA virus infection, limited viral replication, and increased survival.

## Deletion of Med23 promotes the expression of cytokines and ISGs

To evaluate the transcriptome-wide role of Med23 in VSV-induced signaling pathways, we isolated total RNA from Med23$^{\Delta M}$ BMDMs and their controls with or without VSV infection and conducted RNA sequencing (RNA-seq) (S3A Fig).

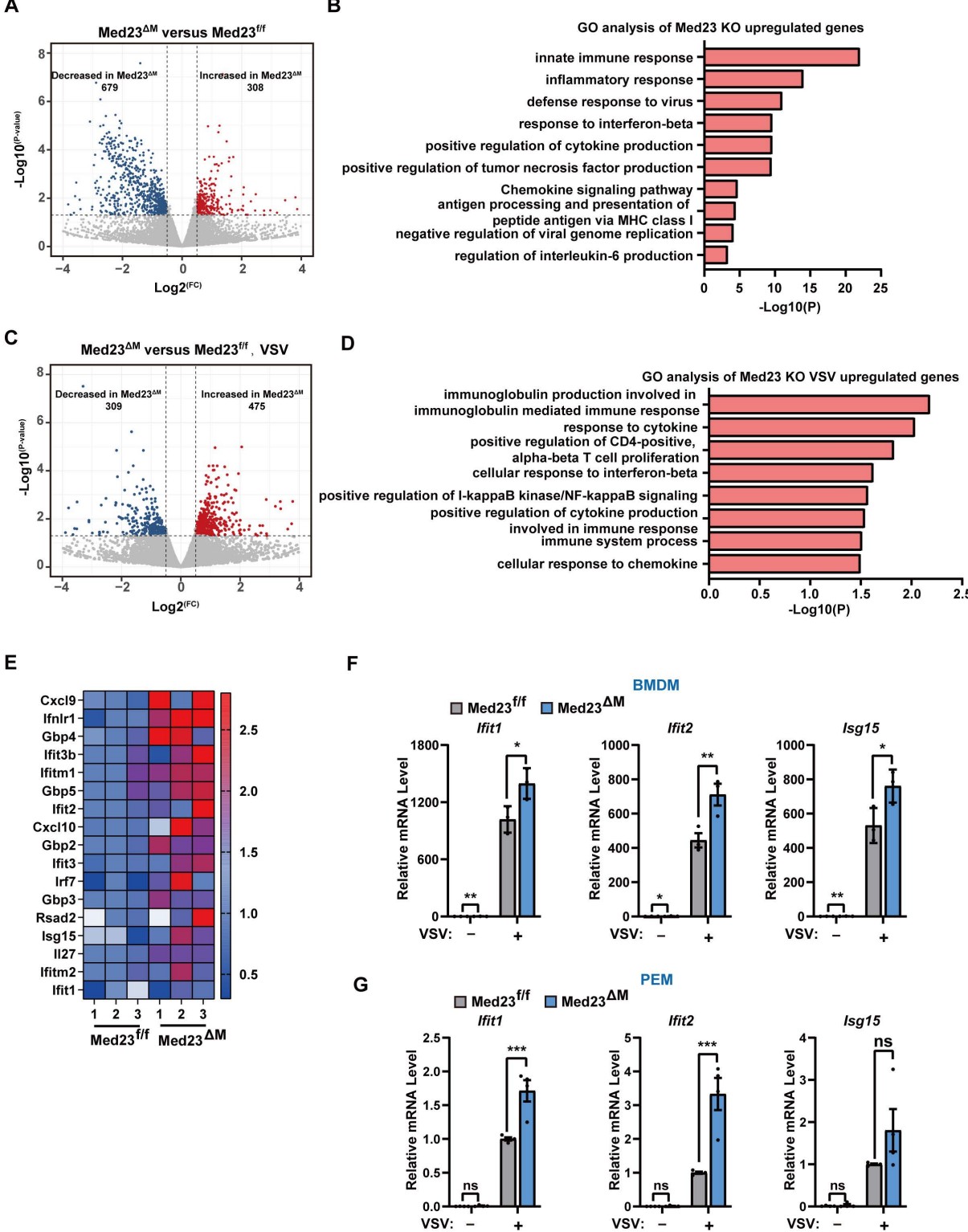

**Fig 4. Increased antiviral innate immune responses in Med23-deficient macrophages. (A)** Volcano plot of genes differentially expressed (false discovery rate < 0.05) in BMDMs according to the RNA-seq data of Med23$^{f/f}$ mice compared with those of Med23$^{\Delta M}$ mice ($n = 3$) (the gray dots represent genes with no statistically significant differences). **(B)** Gene ontology analysis of genes whose expression was increased in BMDMs from Med23$^{\Delta M}$ mice

compared with those from Med23$^{f/f}$ mice ($n = 3$) according to RNA-seq data. **(C)** Volcano plot of genes differentially expressed (false discovery rate < 0.05) in BMDMs infected with VSV (MOI = 0.5) for 4 h according to the RNA-seq data of Med23$^{f/f}$ mice compared with those of Med23$^{ΔM}$ mice ($n = 3$) (the gray dots represent genes with no statistically significant differences). **(D)** Gene ontology analysis of increased genes in BMDMs infected with VSV from Med23$^{ΔM}$ mice compared with those from Med23$^{f/f}$ mice according to RNA-seq data ($n = 3$). **(E)** Heatmap showing the relative expression values (Z score) of differentially expressed ISGs according to the RNA-seq data, ranked by fold change. **(F)** and **(G)** BMDMs ($n = 3$) or PEMs ($n = 4$) from Med23$^{f/f}$ and Med23$^{ΔM}$ mice were stimulated with VSV (MOI = 0.5) for 4 h, and the cells were collected for RT-qPCR analysis of the mRNA expression of Ifit1, Ifit2, and Isg15. **e** Expression levels were normalized to GAPDH mRNA expression and then to the Med23$^{f/f}$ BMDMs without VSV infection. **(G)** Expression levels were normalized to GAPDH mRNA expression and then to the Med23$^{f/f}$ PEMs with VSV infection. Data present means ± SEM of at least three independent experiments. Statistical analysis in (F and G) was performed by two-tailed unpaired Student's $t$ test; ns, not significant, *$P < 0.05$, **$P < 0.01$, and ***$P < 0.001$. The data underlying this figure can be found in S1 Data.

Transcriptome sequencing revealed 306 upregulated and 679 downregulated genes in Med23$^{ΔM}$ BMDMs compared with controls (Fig 4A). GO enrichment analysis revealed that the significantly upregulated genes were involved mainly in the innate immune response, the inflammatory response, the defense response to viruses, and the response to viruses following Med23 depletion (Fig 4B). In the Med23$^{ΔM}$ BMDMs, 475 upregulated and 309 downregulated genes were detected after VSV infection (Fig 4C). These genes whose expression was upregulated in Med23$^{ΔM}$ BMDMs stimulated with VSV were enriched in response to cytokines, the cellular response to interferon-beta, and positive regulation of I-kB kinase/NF-kB signaling (Fig 4D). GO enrichment analysis also revealed that the most highly enriched down-regulated pathways in BMDMs from Med23$^{ΔM}$ mice were associated with cell cycle and DNA replication, such as "regulation of cell cycle process", "activation of the pre-replicative complex," and "DNA replication" (S3B Fig). The sequencing data were further analyzed to verify the increased expression of genes involved in the immune response in Med23$^{ΔM}$ BMDMs, with the mRNA expression of cytokines such as Cxcl9, Cxcl10 elevated compared with that in Med23$^{f/f}$ BMDMs without VSV infection (Fig 4E). Additionally, several ISGs, such as Rsad2, Gbp, and Ifit families, showed increased mRNA levels in Med23$^{ΔM}$ BMDMs. ISGs such as Ifit1, Ifit2, and Isg15 exhibited significantly upregulated mRNA levels in Med23$^{ΔM}$ BMDMs and PEMs infected with VSV (Fig 4F and 4G). Similarly, we analyzed wild-type and Med23 knockout MEFs after VSV infection via RNA-seq (S3C Fig). GO analysis revealed that the most highly upregulated pathways were associated with the response to the virus, particularly the IFN-ISG pathways, such as "defense response to the virus," "cellular response to interferon-beta," and "cytokine-mediated signaling pathway" (S3D Fig). We also measured the expression levels of ISGs in MEFs, where ISGs were significantly induced in Med23-knockout cells after VSV infection (S3E and S3F Fig). The promoters of ISGs contain interferon-stimulated response elements (ISREs) [29]. We next used an ISRE luciferase reporter to evaluate the regulatory effect of Med23 on ISGs through VSV infection or poly(I:C) stimulation. As shown in S3G and S3H Fig, ISRE luciferase activity was significantly induced in Med23-deficient HeLa cells. Furthermore, re-expression of the Med23 protein in Med23 KO cells markedly inhibited VSV- or poly(I:C)-induced ISRE luciferase activity (S3I Fig). Overall, these results demonstrate that Med23-deficiency enhances the innate immune signaling, resulting in increased ISGs expression during VSV infection.

## Med23 deficiency transcriptionally upregulates RIG-I to increase type I interferon production

We next sought to determine which targets could be directly involved in the upregulation of the IFN response caused by Med23 deficiency. HeLa cells were transfected with plasmids expressing MAVS, TBK1, or IRF3, which can turn on the transcription of IFNβ. We observed that the presence or absence of Med23 did not alter IFNβ mRNA levels in response to MAVS, TBK1, or IRF3 overexpression, suggesting that Med23 targets an upstream component of the RLR signaling pathway (Fig 5A). We also observed a significant increase in the protein level of RIG-I, a cytosolic RNA sensor crucial for activating the immune response, in Med23$^{ΔM}$ BMDMs following VSV infection (Fig 5B). Consistently, the RIG-I protein was more abundant in Med23 knockout MEFs and RAW264.7 cells after VSV infection or poly(I:C) stimulation (S4A–S4C Fig).

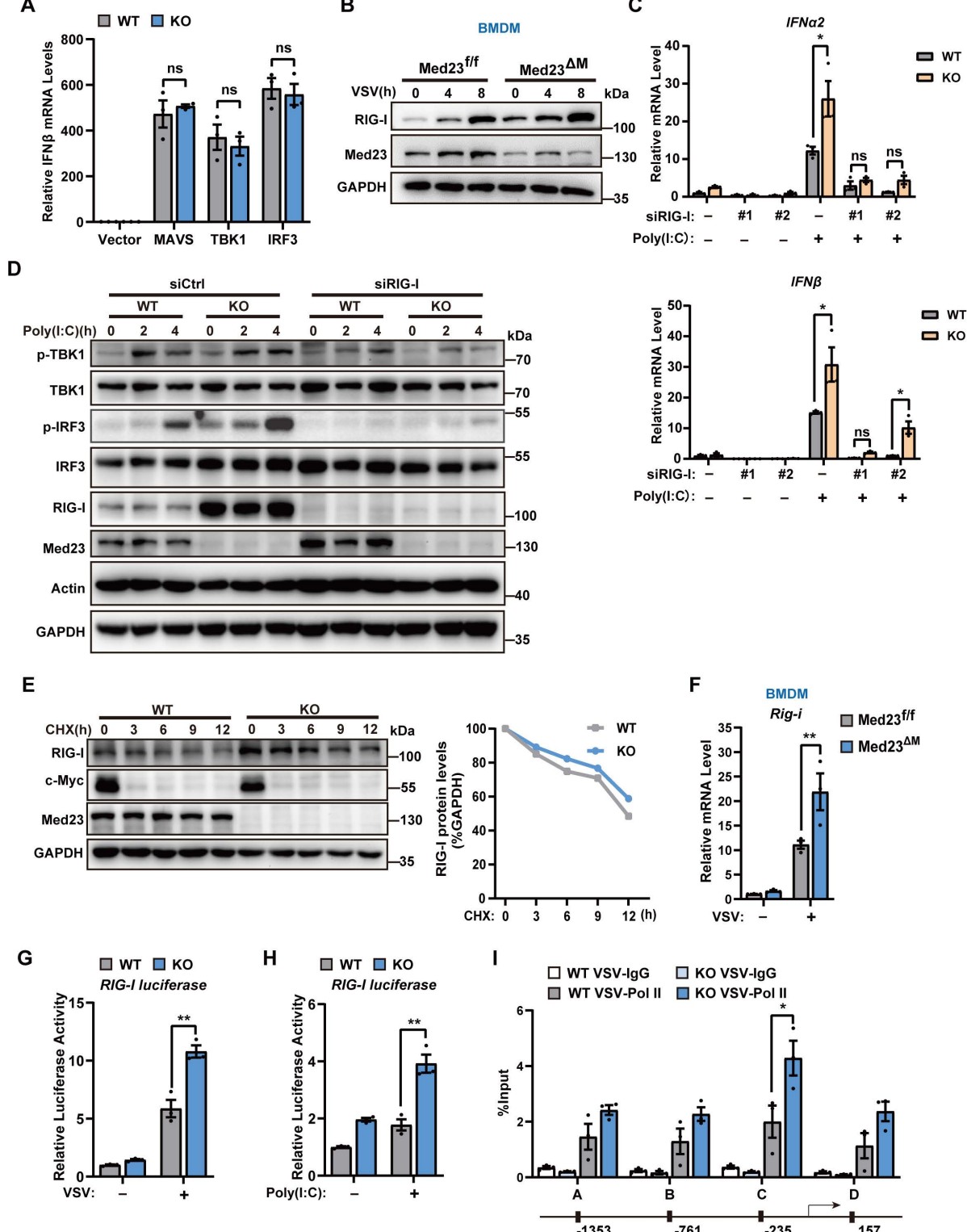

**Fig 5. Med23 deficiency enhances the transcription of RIG-I, leading to increased production of type I interferons. (A)** The expression of IFNβ mRNA in WT and Med23 knockout MEFs (*n* = 3) transfected with expression plasmids for MAVS, TBK1, and IRF3 or empty control vector for 48 h. The results are presented relative to the expression of IFNβ mRNA in control cells. **(B)** Immunoblot analysis of RIG-I in BMDMs from Med23^f/f^ and Med23^ΔM^

mice. **(C)** Expression of IFNα2 and IFNβ mRNAs in WT and Med23 knockout MEFs ($n=3$) with RIG-I knockdown (siRNA = 50 nM) for 48 h and then stimulated with poly(I:C) (5 µg/mL) for 8 h. **(D)** Immunoblot analysis of the indicated protein in WT and Med23 knockout MEFs with RIG-I knockdown (siRNA = 50 nM) for 48 h and then stimulated with poly(I:C) (5 µg/mL) for indicated time. **(E)** Immunoblot analysis of RIG-I in WT and Med23 knockout HeLa cells ($n=3$) treated with CHX (50 ng/mL). GAPDH served as a loading control. The relative RIG-I protein levels are shown in the right panel. The protein density was quantified via ImageJ software. **(F)** Expression of RIG-I mRNA in BMDMs from Med23$^{f/f}$ and Med23$^{ΔM}$ mice ($n=3$). **(G)** and **(H)** Luciferase activity of the RIG-I promoter-reporter in WT and Med23 knockout HeLa cells ($n=3$) unstimulated or stimulated with VSV (MOI = 0.5) or poly(I:C) (5 µg/mL). **(I)** ChIP-qPCR analysis of Pol II-bound sites in the RIG-I promoter in WT and Med23-knockout Raw264.7 cells ($n=3$) infected with VSV (MOI = 0.5). Data present means ± SEM of three independent experiments. Statistical analysis in (A), (C), and (F–I) was performed by two-tailed unpaired Student $t$ test; ns, not significant, *$P < 0.05$ and **$P < 0.01$. The data underlying this figure can be found in S1 Data.

Conversely, exogenous Med23 expression in Med23 knockout MEFs reduced RIG-I levels after VSV stimulation (S4D Fig), suggesting that RIG-I is a possible effector of Med23 in antiviral signaling.

To investigate whether the enhanced innate immune response in Med23-deficient cells depends on RIG-I, we knocked down RIG-I with siRNAs in wild-type and Med23 knockout MEFs and challenged them with VSV, followed by detection of IFNβ and IFNα2 mRNA expression. Importantly, the VSV-stimulated increase in the expression of IFNβ and IFNα2 in both WT and Med23-knockout cells was largely abrogated by RIG-I knockdown (Fig 5C). Moreover, RIG-I knockdown significantly blunted the Med23 knockout-induced increase in the phosphorylation of TBK1 and IRF3 (Fig 5D). These results suggest that Med23-mediated suppression of innate immune responses against RNA viruses apparently relies on RIG-I.

To determine whether RIG-I protein accumulation in KO cells was due to inhibited protein degradation, we treated wild-type and Med23 knockout HeLa cells with cycloheximide (CHX), a protein biosynthesis inhibitor, and monitored RIG-I protein levels for the indicated times. The RIG-I protein underwent significant degradation after CHX treatment in both wild-type and KO HeLa cells, and Med23 deficiency did not affect the half-life of the endogenous RIG-I protein (Fig 5E). Furthermore, RIG-I mRNA stability remains unchanged in both wild-type and Med23 knockout cells following treatment with transcriptional inhibitor Actinomycin D (Act D), suggesting that Med23 may regulate RIG-I at the transcriptional level (S4E Fig). We then tested whether Med23 influences RIG-I transcription and quantified RIG-I mRNA levels in Med23$^{f/f}$ and Med23$^{ΔM}$ BMDMs after VSV infection, which revealed higher levels in Med23$^{ΔM}$ cells (Fig 5F). Similarly, RIG-I mRNA was elevated in Med23 knockout cells compared with WT MEFs infected with VSV or stimulated with poly(I:C) (S4F and S4G Fig). In contrast, exogenous Med23 expression in Med23 knockout MEFs led to decreased RIG-I mRNA level (S4H Fig), suggesting that Med23 may transcriptionally control the expression of RIG-I.

To verify whether Med23 directly controls RIG-I at the gene expression level, dual-luciferase reporter assays were conducted to evaluate the impact of Med23 on RIG-I promoter activity. Med23 deletion increased RIG-I promoter activity following VSV infection or poly(I:C) stimulation (Fig 5G and 5H). ChIP assays using a Pol II antibody in RAW264.7 cells infected with VSV revealed increased Pol II recruitment to RIG-I promoters in the absence of Med23 (Fig 5I). Therefore, our data suggest that Med23 may act as a corepressor in the transcriptional regulation of RIG-I to block IFN-I.

## Med23 cooperates with Foxo3 to suppress RIG-I expression

RIG-I is an ISG induced by IFN-I. To investigate whether the observed effects are due to Med23-mediated transcriptional regulation of RIG-I or an indirect effect via IFN-mediated induction, we knocked down IFNAR1 using siRNAs in wild-type and Med23 knockout RAW264.7 cells and challenged them with VSV, followed by analysis of RIG-I mRNA expression. Importantly, the VSV-induced upregulation of RIG-I expression in both WT and Med23-knockout cells was partially reduced by IFNAR1 knockdown. Moreover, Med23 depletion still significantly induced RIG-I mRNA levels in IFNAR1-knockdown cells, suggesting a IFN-independent, constitutive regulatory mechanism (S5A Fig). Given that Med23 controls RIG-I expression at the transcriptional level, we attempted to determine which transcription factor might collaborate with Med23 to regulate RIG-I expression. Using EPD (https://epd.epfl.ch), a resource for predicting transcription factor targets, we identified Foxo3 as a potential transcription factor that binds to the RIG-I promoter (S5B Fig). Previous studies have implicated Foxo3 in the regulation of

viral replication [26,27]. We observed that Foxo3 overexpression significantly reduced RIG-I promoter-driven luciferase activity in a dose-dependent manner (Fig 6A), but compared with control MEFs, Foxo3-specific siRNAs resulted in increased RIG-I expression levels (Fig 6B). Moreover, Foxo3 knockdown increased RIG-I mRNA and protein levels after VSV infection (Fig 6C and 6D), which phenocopied the effect of Med23 deficiency. We further examined the impact of Foxo3 on the IFN-I response to VSV stimulation and found that similar to Med23 KO, Foxo3 knockdown significantly increased VSV-triggered IFNβ, IFNα2, and Ifit1 mRNA expression (Fig 6E). Together, Foxo3 suppresses RIG-I expression to block IFN-I production.

As Med23 and Foxo3 act similarly in controlling VSV-induced gene expression, we performed coimmunoprecipitation experiments and observed that Myc-Med23 physically associates with exogenous FLAG-Foxo3 (Fig 6F). Importantly, endogenous HA-Med23 interacts not only with Mediator subunits Med1, Med16, Med24, and CDK8 but also associates with Foxo3 (Fig 6F). We further performed the DSP crosslinking-coupled IP experiment, an approach to distinguish direct from indirect interactions. The interaction between Med23 and Foxo3 was retained following DSP crosslinking, supporting a direct physical interaction between Med23 and Foxo3 (S5C Fig). We carried out a ChIP-qPCR assay and detected reduced Foxo3 binding to the RIG-I promoter region in VSV-infected Med23 KO RAW264.7 cells compared with WT cells (Fig 6G). We further analyzed the functional interaction between Med23 and Foxo3 at the genome-wide scale through a comparison of our RNA-seq data with previously identified Foxo3 targets [26] (S5D Fig). This analysis revealed that a large fraction of Foxo3 target genes were upregulated in Med23 knockout cells (Fig 6H). Together, these results indicate that Med23 cooperates with Foxo3 to repress RIG-I expression.

## Discussion

In this study, we demonstrated that the Mediator subunit Med23 cooperates with Foxo3 to repress the transcription of the cytosolic viral RNA receptor RIG-I, which serves as a checkpoint for preventing the hyper-antiviral response and viral replication (Fig 7). These findings demonstrate that Med23 is essential for controlling viral infection and suppressing host innate immunity.

Med23 has been identified as a crucial regulator in various biological processes; however, its role in antiviral innate immune signaling remains unknown. Previous studies have reported that the Mediator complex is crucial for DNA virus HSV-1 replication, as depletion of multiple Mediator subunits significantly reduces virus replication in cells [30]. Additionally, the loss of the Mediator tail module enhances cell survival against SARS-CoV-2 and pan coronavirus infections in Huh7.5 cells [31]. However, the mechanism underlying the action of Mediators in antiviral innate immunity is unknown, and in vivo evidence is needed. In our study, we found that Mediator tail subunit Med23 deficiency enhanced IFN-I production induced by the RNA virus VSV, thereby combating viral replication. Moreover, we generated macrophage-conditional Med23 knockout mice and demonstrated that compared with Med23f/f macrophages, primary macrophages with Med23 deficiency presented significantly increased IFN-I mRNA levels. Notably, Med23-deficient mice presented increased resistance to VSV infection, with reduced weight loss following low-dose challenge, lower mortality upon high-dose exposure, and a reduced incidence of VSV-induced pathological lesions in the lung in vivo. This report shows that Med23 deficiency in macrophages facilitates the IFN-I signaling pathway, thus suppressing RNA viral replication in vivo.

Mechanistically, this study revealed that Med23 negatively regulates the transcription of RIG-I, a receptor protein essential for recognizing cytoplasmic viral RNA and activating the IFN-I signaling pathway [32]. RIG-I-deficient cells and mice are more vulnerable to infections by VSV, influenza A virus, hepatitis C virus, and Japanese encephalitis virus, highlighting the crucial role of RIG-I in the IFN-I pathway [33,34]. RIG-I-mediated antiviral innate immune signaling is regulated by many host proteins at the posttranslational modification level. For example, NLRP12 interacts with the ubiquitin ligase TRIM25 to prevent K63-linked ubiquitination of RIG-I [35]; Siglec-G recruits SHP2 and the E3 ubiquitin ligase c-Cbl to RIG-I, leading to its degradation via K48-linked ubiquitination [36]. Nevertheless, less is known about the role of nuclear molecules in the innate antiviral immune response at the transcriptional level. Our data showed that Med23

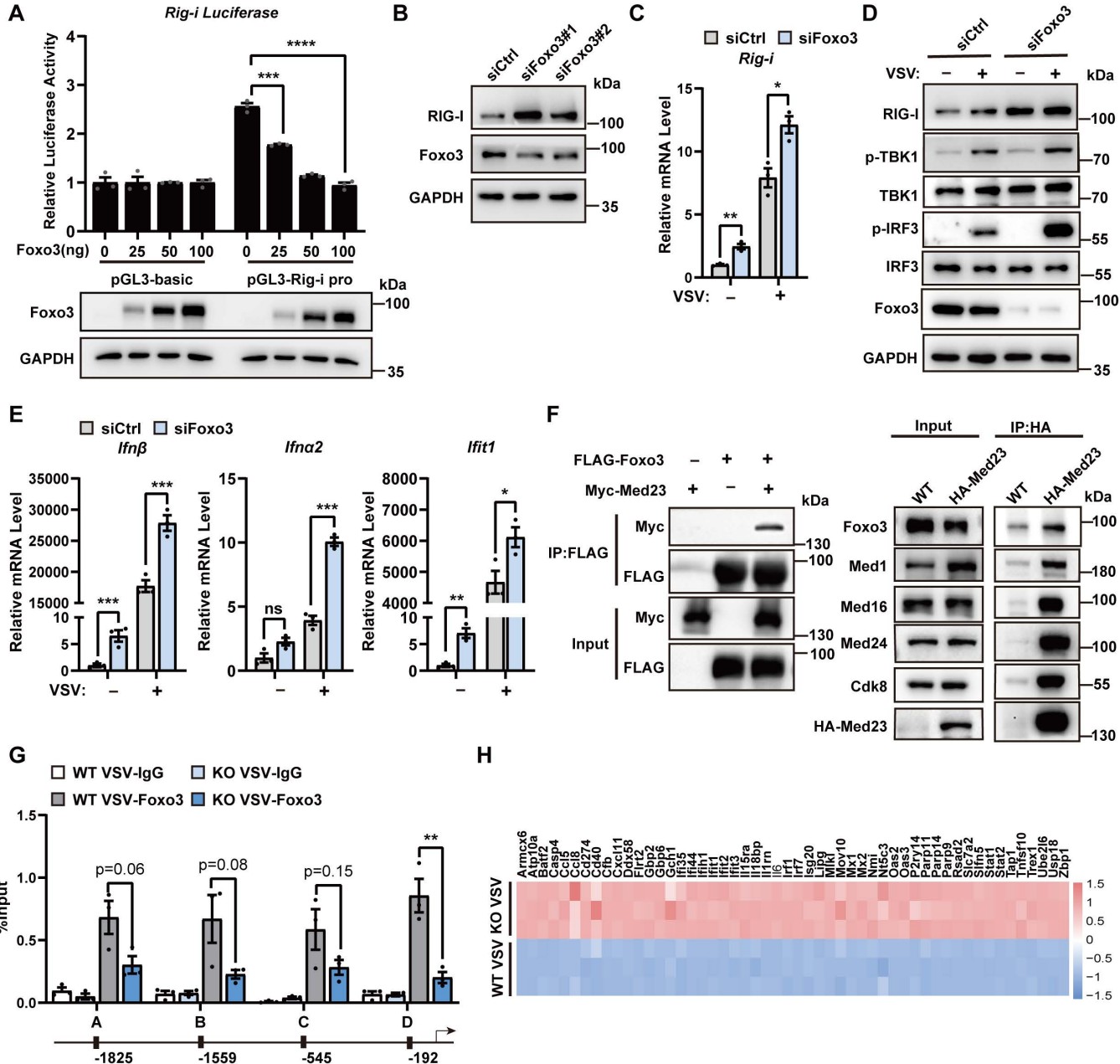

**Fig 6. Med23 cooperates with FOXO3 to negatively regulate RIG-I. (A)** Luciferase activity assay in MEF cells ($n = 3$) conducted 48 h after co-transfected with gradient doses of the Foxo3 plasmid with or without the RIG-I promoter reporter plasmid (upper panel); immunoblot analysis of Foxo3 overexpression in HeLa cells (lower panel). **(B)** Immunoblot analysis of RIG-I and Foxo3 in Foxo3-knockdown Raw264.7 cells (siRNA = 50nM). **(C)** qPCR analysis of Rig-i mRNA levels in Foxo3-knockdown RAW264.7 cells ($n = 3$) infected with VSV (MOI = 0.5) for 6 h. **e** Immunoblot analysis of the indicated proteins in Foxo3-knockdown RAW264.7 cells infected with VSV(MOI = 0.5) for 6 h. **(E)** qPCR analysis of Ifnβ, Ifnα2, and Ifit1 expression in Foxo3-knockdown RAW264.7 cells ($n = 3$) infected with VSV (MOI = 0.5) for 6 h. **(F)** Immunoblot analysis of the interaction between tagged Foxo3 and Med23 in HEK293T cell lysates immunoprecipitated with the FLAG antibody (left panel); immunoblot analysis of endogenous Foxo3 in the cell lysates of HA-Med23-KI HEK293T cells immunoprecipitated with the HA antibody (right panel). **(G)** ChIP-qPCR analysis of Foxo3-bound sites in RIG-I promoters with VSV infection ($n = 3$, MOI = 0.5). **(H)** Heatmap analysis revealed the expression profiles of downstream target genes regulated by Foxo3 via the RNA-Seq data of Med23 knockout MEFs. Data present means ± SEM of three independent experiments. Statistical analysis in (A), (C), (E), and (G) was performed by two-tailed unpaired Student $t$ test; *$P < 0.05$, **$P < 0.01$, and ***$P < 0.001$. The data underlying this figure can be found in S1 Data.

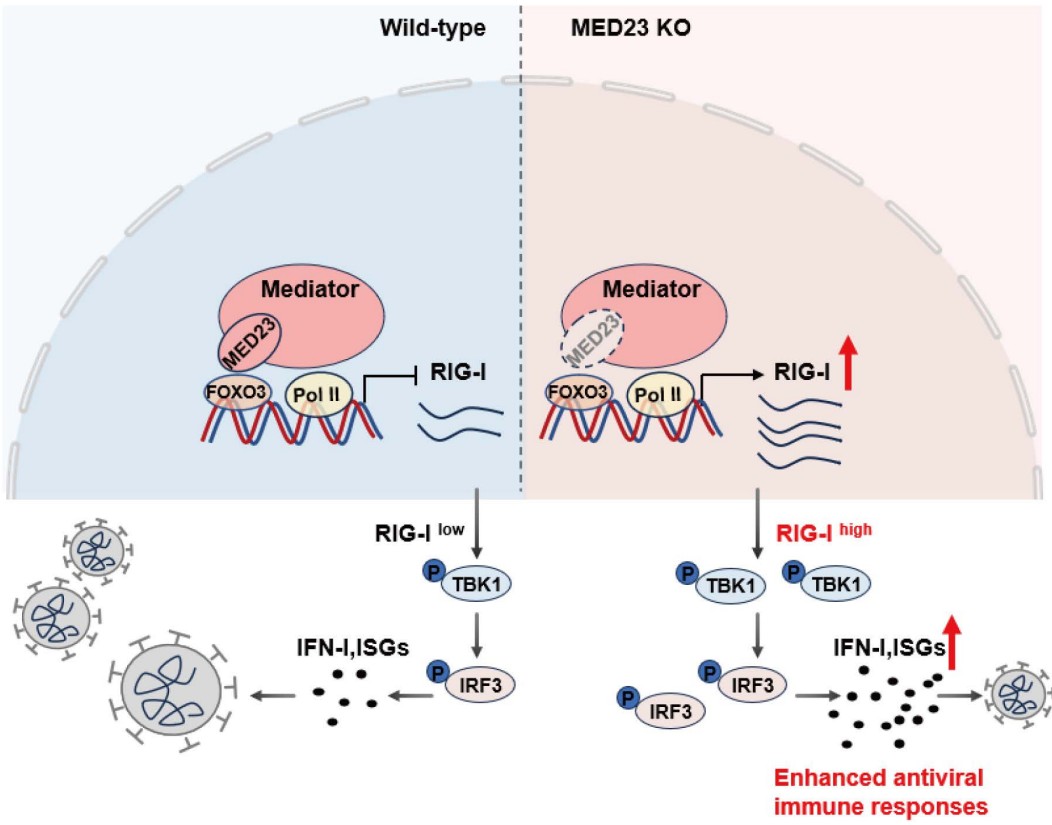

**Fig 7. Schematic showing that Med23 inhibits the antiviral IFN-I response by negatively regulating RIG-I.** Med23 interacts with Foxo3 to transcriptionally repress RIG-I, thereby inhibiting the RNA virus-induced phosphorylation of TBK1 and IRF3, suppressing antiviral innate immune response, and promoting virus replication.

transcriptionally regulates RIG-I expression, restoring IFN-I signaling following viral infection. However, how Med23 supports the maintenance of homeostasis in IFN signaling remains uncertain.

Although the Mediator complex is widely established as a transcriptional coactivator, emerging evidence from multiple studies indicates that specific Mediator subunits can also exert gene-repressive functions [24,37–40]. For example, MED19 and MED26 act as molecular scaffolds linking the RE1-silencing transcription factor to the G9a histone methyltransferase, thereby suppressing neuronal gene expression via histone H3K9 methylation in non-neuronal cells [40]. In our investigation into the mechanistic role of Med23 in regulating RIG-I, we found that Med23 suppresses RIG-I expression through Foxo3-mediated transcriptional repression. Foxo3, a forkhead box family transcription factor, is known to negatively regulate the antiviral response [26] and is highly expressed in bone marrow [41]. Our data suggest that Med23 physically interacts with Foxo3 to inhibit RIG-I function during the immune response. Notably, prior research has shown that Foxo3 protein undergoes stress-induced degradation [26], raising the possibility that viral infection may disrupt the Med23-Foxo3 interaction. Additionally, MED23 may modulate the activity of other transcriptional cofactors, thereby influencing the cooperative binding of Foxo3 to the RIG-I promoter. This is analogous to recent findings showing that FOXA1 binding to chromatin is profoundly influenced by co-transcription factors such as AP-1 and CEBPB [42]. Furthermore, Foxo3 is known to repress gene expression by forming a trimeric complex with nuclear corepressor 2 and histone deacetylase 3, leading to a reduction in histone

H4 acetylation on target gene promoters [26]. We hypothesize that Foxo3 may repress RIG-I expression through a similar mechanism, whereby MED23 facilitates its association with repressive complexes or corepressors. Future studies will aim to characterize the dynamic regulatory network in which Med23 participates during antiviral signaling and to define the precise molecular mechanisms underlying its role in modulating antiviral immune responses.

In conclusion, this work demonstrated that Med23 potentiates the transcription of RIG-I, thereby counteracting the innate immune response and facilitating viral replication. Our findings indicate that Med23 associates with Foxo3 and inhibits RIG-I transcription following RNA virus infection, ultimately suppressing the antiviral response. This study revealed that Med23 is essential for viral infection and the suppression of host innate immunity, suggesting that it could serve as a potential target for the development of antiviral therapeutics.

## Materials and methods

### Ethics statement

All animal experiments in this study were performed in accordance with the institutional ethical guidelines for animal experiments and were approved by the Fudan University Animal Care and Use Committee (Animal ethical license number:2025JS016).

### Mice

Med23-floxed mice (Med23$^{f/f}$) were generated by homologous recombination as previous described [22]. LysM-Cre mice were generously provided by Dr. Hongyan Wang at Shanghai Institute of Biochemistry and Cell Biology. Med23$^{f/f}$ mice were bred to LysM-Cre mice to generate myeloid-specific Med23 knockout mice (termed Med23$^{\Delta M}$). All animal experiments used 6- to 8-week-old, age- and sex-matched mice in C57BL/6 background.

### Cell culture and preparation of macrophages

MED23 knockout HeLa and RAW264.7 cell lines were generated using the CRISPR/Cas9 plasmid vector PX330, and single-cell clones were subsequently isolated by FACS. MEF, RAW264.7, HeLa, and 293T cell lines were maintained in complete DMEM with 10% FBS and 1% penicillin-streptomycin. Bone marrow cells were isolated from mouse femurs and cultured at 37 °C in RPMI1640 containing 10% FBS and 1% penicillin-streptomycin, with M-CSF (10 ng/ml, Peprotech) for BMDM differentiation [43]. The medium was changed every 3 days. On day 7, the cells were used for analysis. Mouse PEMs were harvested 3 days after intraperitoneal injection of 2 mL of 3% Brewer thioglycolate medium and cultured in RPMI1640 with 10% FBS and 1% penicillin-streptomycin [44].

### Dual-Luciferase reporter assay

The human or mouse RIG-I promoter region (2 kb upstream of TSS) was cloned into the pGL3-Basic vector upstream of the firefly luciferase gene. The luciferase reporter assay was conducted as previously described [45]. Briefly, 293T and HeLa cells were seeded at $1 \times 10^5$ cells per well in a 12-well plate and incubated overnight. The cells were then transfected using Lipo2000 with a luciferase reporter plasmid, pRL-TK, and the indicated plasmids. After 36–48 h, the cells were harvested and analyzed using a dual-luciferase reporter assay system (Promega) following the manufacturer's instructions.

### Immunoprecipitation and Western blotting analysis

293T cells were co-transfected with the appropriate plasmids. After 48 h, the cells were harvested and lysed in a buffer containing 0.2% Tween-20, 150 mM NaCl, 1 mM EDTA, 25 mM EGTA, and 50 mM HEPES (pH 7.5), with added protease

inhibitors. The lysates were subjected to immunoprecipitation with antibodies overnight at 4 °C, followed by washing in lysis buffer. Equalized extracts were used for immunoprecipitation and Western blotting analysis with the indicated antibody.

## RNA isolation and quantitative reverse transcription PCR (qRT-PCR)

Total RNA was extracted from mouse tissues or cells using TRIzol reagent (Invitrogen) and reverse transcribed with reverse transcriptase (Vazyme). The resulting cDNA samples were amplified by quantitative PCR with SYBR Green Real-time PCR Master Mix (Vazyme) and gene-specific primers using a Life QuantStudio7 Flex Real-Time PCR System. The relative transcript levels were normalized to GAPDH. Primer sequences are listed in S1 Table.

## Hematoxylin and eosin (H&E) staining

For histological analysis, lung tissues fixed in 4% paraformaldehyde for 24 hours were dehydrated, embedded in paraffin, and sliced into 5-mm sections. These sections were dewaxed, stained with H&E, and examined under a light microscope. The injury score was calculated based on alveolar wall thickening, fibroblast proliferation, and extracellular matrix deposition, as described in previous studies [46].

## Flow cytometry

Spleen single-cell suspensions were prepared, and RBCs were lysed before staining. The cell suspensions were divided into two parts. One part was stained with antibodies specific for mouse CD45 (APC), CD3 (APC-Cy7), CD4 (FITC), CD8a (PE), CD19 (BV421), and NK1.1 (PE-Cy7). The other part was stained with antibodies for mouse CD45 (APC), MHCII (AF700), F4/80 (BV421), CD11b (PerCP-Cy5.5), CD11c (FITC), and Ly6G (PE). After incubating in the dark for 20 min, cells were washed three times and then analyzed by FACS. Cells were infected with VSV-GFP, and the level of green fluorescence was detected using a FACS cytometer (Fortessa, BD). Data were analyzed with FlowJo software (TreeStar).

## RNA-seq and data analysis

RNA was extracted using TRIzol Reagent. The purity and integrity of the RNA were confirmed with an Agilent Bioanalyzer. RNA-seq library preparation and sequencing were performed on an Illumina 2500 machine. After quality assessment of the FASTQ files with FastQC, adapter sequences were trimmed from the reads. Read alignment and junction finding were completed using TopHat, and differential gene expression was analyzed with Cuffdiff, using the UCSC mm10 assembly as the reference sequence. Genes with a |fold change| > 1.5 and a false discovery rate-adjusted $P$-value ≤0.05 were selected for further analysis. Gene Ontology (GO) analysis of differentially expressed genes was performed using the Metascape online tool. The RNA-seq data have been deposited in the Gene Expression Omnibus (GEO) database.

## ChIP

The cells were cross-linked with 1% formaldehyde for 10 min at room temperature, and the cross-linking was quenched with 0.125 M glycine for 5 min. After washing twice with precooled PBS, the cells were homogenized in ChIP lysis buffer (50 mM Tris-HCl, pH 7.4; 1% SDS; and 10 mM EDTA). The pellets were then sonicated to obtain chromatin fragments between 200 and 500 bp in length. The subsequent steps followed previously described protocols [47]. The antibodies used for ChIP were: anti-Pol II (Santa Cruz, B240764) and anti-FOXO3 (CST, 12,829). ChIP-qPCR primers (listed in S2 Table) were designed to target Pol II-binding regions in the RIG-I promoter based on ENCODE ChIP-seq data and predicted FOXO3-binding motifs based on JASPAR.

## Statistical analysis

All data are presented as means ± SEM. Statistical significance between data sets was determined using a two-tailed unpaired Student's $t$ test or Mann–Whitney test with GraphPad Prism 8. Differences were considered statistically significant at $P \leq 0.05$ (*$P < 0.05$, **$P < 0.01$, ***$P < 0.001$, ****$P < 0.0001$; ns, not statistically significant).

## Supporting information

**S1 Fig. Increased expression of type I interferons and proinflammatory cytokines after Med23 knockout. (A)** Immunoblotting to detect Med23 in wild-type and Med23-knockout MEF cells. **(B)** and **(C)** ELISA analysis of IFNβ in the supernatant of Med23-overexpressing KO MEFs ($n = 4$) infected with VSV (MOI = 1) or stimulated with poly(I:C) (5 μg/mL) for 8 h. **(D)** Immunoblotting was used to detect Med23 in wild-type and Med23-knockout RAW264.7 cells. **(E)** qPCR analysis of the levels of the indicated genes in Med23 knockout RAW264.7 cells infected with VSV (MOI = 1) for 6 h. **(F)** Immunoblotting to detect Med23 in wild-type and Med23-knockout HeLa cells. **(G)** and **(H)** qPCR analysis of the levels of the indicated genes in Med23-knockout HeLa cells ($n = 3$) infected with VSV (MOI = 1) or stimulated with poly(I:C) (5 μg/mL) for the indicated times. **(I)** Med23 knockout HeLa cells ($n = 3$) were transiently transfected with IFNβ reporter plasmids along with Med23-expressing plasmids (0, 100, 200, and 300 ng). After 48 h, a luciferase assay was performed in which the cells were stimulated with VSV (MOI = 1) for 8 h. The results are presented relative to the luciferase activity in control cells (transfected with the luciferase reporter and empty vector with stimulation of VSV). The results of the immunoblot analysis of Med23 are shown below. Actin and GAPDH served as loading controls. **(J)** Immunoblot analysis of the indicated protein in Med23 knockout RAW264.7 cells infected with VSV (MOI = 1). **(K)** and **(L)** Immunoblot analysis of the indicated protein in Med23-knockout HeLa cells infected with VSV (MOI = 1) or stimulated with poly(I:C) (5 μg/mL). **(M)** Immunoblot analysis of p-IRF3 protein in the nuclear and cytoplasmic fractions of WT and Med23-knockout HeLa cells upon stimulation with poly(I:C) (5 μg/mL) for 8 h. GAPDH served as a cytoplasmic control. Lamin B1 served as a nuclear protein control. **(N)** TCID$_{50}$ assay of virus titers of WT and Med23 KO MEFs ($n = 3$) after VSV infection (MOI = 1). **(O)** TCID$_{50}$ assay of virus titers of Med23-overexpressing KO MEFs ($n = 3$) after VSV infection (MOI = 1) for 8 h. **(P)** Flow cytometry analysis of GFP fluorescence intensity in WT and Med23-knockout RAW264.7 cells infected with VSV-GFP (MOI = 1) for 8 h. **(Q)** Flow cytometry analysis of GFP fluorescence intensity in WT and Med23-knockout HeLa cells infected with VSV-GFP (MOI = 1) for 8 h. Data present means ± SEM of three independent experiments. Statistical analysis in (B–C), (E), (G–I), and (N–Q) was performed by two-tailed unpaired Student $t$ test; *$P < 0.05$, **$P < 0.01$, and ***$P < 0.001$. The data underlying this figure can be found in S2 Data.
(TIF)

**S2 Fig. The number of immune cells was comparable between Med23$^{f/f}$ mice and Med23$^{\Delta M}$ mice. (A)** Strategy for generation of macrophage-specific Med23 conditional knockout mice. **(B)** qPCR analyses of Med23 gene expression in BMDMs and PMs from Med23$^{f/f}$ and Med23$^{\Delta M}$ mice ($n = 3$). **(C)** and **(D)** Flow cytometric analysis of the percentages of BMDMs or PEMs from Med23$^{f/f}$ and Med23$^{\Delta M}$ mice ($n = 3$). **(E–I)** Flow cytometric analysis of B cells, CD4+ T cells, CD8+ T cells, NK cells, neutrophils, dendritic cells and macrophages in the spleens of Med23$^{f/f}$ and Med23$^{\Delta M}$ mice ($n = 3$). **(J)** Statistical analysis of the percentages of immune cells in E–I. Data present means ± SEM of three independent experiments. Statistical analysis in (B–C), and (J) was performed by two-tailed unpaired Student $t$ test; ns, not significant. The data underlying this figure can be found in S2 Data.
(TIF)

**S3 Fig. Increased antiviral innate immune responses in Med23-deficient cells. (A)** Cluster heatmap of the RNA-seq data showing the differentially expressed genes in the Med23$^{f/f}$ and Med23$^{\Delta M}$ BMDMs without or with VSV infection. **(B)** Gene ontology analysis of down-regulated genes the Med23$^{f/f}$ and Med23$^{\Delta M}$ BMDMs without or with VSV infection

according to RNA-seq data. **(C)** Cluster heatmap of the RNA-seq data showing the differentially expressed genes in the wild-type and Med23-knockout MEFs without or with VSV infection. **(D)** Gene ontology analysis of increased genes in WT and Med23 knockout MEFs infected with VSV according to RNA-seq data. **(E)** and **(F)** qRT-PCR analysis of ISGs in WT and Med23 knockout MEFs infected with VSV ($n=4$) or stimulated with poly(I:C) ($n=3$) for the indicated times. **(G)** and **(H)** Luciferase reporter assays were conducted with WT and Med23-knockout HeLa cells ($n=3$) transfected for 48 h with ISRE reporter plasmids and then infected with VSV (MOI = 1) or stimulated with poly(I:C) (5 μg/mL) for 8 h. **(I)** Med23 knockout HeLa cells ($n=3$) were transiently transfected with IFNβ reporter plasmids along with Med23-expressing plasmids (0, 100, 200, and 300 ng). After 48 h, a luciferase assay was performed in which the cells were stimulated with VSV (MOI = 1) for 8 h. The results are presented relative to the luciferase activity in control cells (transfected with the luciferase reporter and empty vector with stimulation of VSV). The results of the immunoblot analysis of Med23 are shown below. Data present means ± SEM of at least three independent experiments. Statistical analysis in (E–I) was performed by two-tailed unpaired Student's $t$ test; $*P<0.05$, $**P<0.01$, and $***P<0.001$. The data underlying this figure can be found in S2 Data.
(TIF)

**S4 Fig. Med23 deficiency transcriptionally upregulates RIG-I to increase type I interferon production. (A)** and **(B)** Immunoblot analysis of RIG-I in WT and Med23 knockout MEFs ($n=3$) infected with VSV (MOI = 1) or stimulated with poly(I:C) (5 μg/mL) for the indicated times. **(C)** Immunoblot analysis of RIG-I in WT and Med23 knockout RAW264.7 cells infected with VSV (MOI = 0.5) for the indicated times. **(D)** Immunoblot analysis of RIG-I protein expression in Med23-overexpressing knockout MEFs infected with VSV (MOI = 1) for 8 h. **(E)** qPCR analysis of RIG-I in WT and Med23 knockout HeLa cells ($n=3$) treated with Act D (1 mM/mL) for the indicated times. **(F)** and **(G)** qPCR analysis of RIG-I mRNA levels in WT and Med23 knockout MEFs infected with VSV ($n=4$, MOI = 1) or stimulated with poly(I:C) ($n=3$, 5 μg/mL) for the indicated times. **(H)** qPCR analysis of RIG-I mRNA levels in Med23-overexpressing knockout MEFs ($n=3$) stimulated with poly(I:C) (5 μg/mL) for 8 h. Data present means ± SEM of at least three independent experiments. Statistical analysis in (F–H) was performed by two-tailed unpaired Student $t$ test; $*P<0.05$, $**P<0.01$, and $***P<0.001$. The data underlying this figure can be found in S2 Data.
(TIF)

**S5 Fig. Med23 cooperates with Foxo3 to suppress RIG-I expression. (A)** qPCR analysis of Rig-i and Ifnar1 mRNAs in WT and Med23 knockout MEFs ($n=3$) with Ifnar1 knockdown (siRNA = 50nM) infected with VSV (MOI = 1) for 8 h. **(B)** schematic illustration of the predicted binding sites of Foxo3 on the RIG-I promoter was generated utilizing the resource provided at https://epd.epfl.ch. **(C)** Immunoblot analysis of endogenous Foxo3 in the cell lysates of HA-Med23-KI HEK293T cells treated with DSP (20uM) immunoprecipitated with the HA antibody. **(D)** Venn diagram to show the overlap of these genes upregulated in FOXO3 KO cells treated with or without poly(I:C) and those induced by VSV infection in our RNA-seq analysis. Data present means ± SEM of three independent experiments. Statistical analysis in (A) was performed by two-tailed unpaired Student $t$ test; $*P<0.05$, $**P<0.01$, and $***P<0.001$. The data underlying this figure can be found in S2 Data.
(TIF)

**S1 Table. Primer sequences used in the qRT-PCR analysis.**
(PDF)

**S2 Table. Primer sequences used in the ChIP-qPCR analysis.**
(PDF)

**S1 Data. Numerical data used in** Figs 1–6.
(XLSX)

**S2 Data.   Numerical data used in S1–S5 Figs.**
(XLSX)

**S3 Data.  Gating strategies of FACS.**
(PDF)

**S1 Raw Images.  Original western blots images for Figs.**
(PDF)

**S1 File.  FACS.**
(ZIP)

## Acknowledgments

We thank Dr. Hong-yan Wang (Shanghai Institute of Biochemistry and Cell Biology, China) for providing LysM-Cre mice, Dr. Fa-jian Hou (Shanghai Institute of Biochemistry and Cell Biology, China) for providing VSV, IFN-β, and ISRE luciferase reporter plasmids.

## Author contributions

**Conceptualization:** De-fei Xiong, Gang Wang.

**Data curation:** De-fei Xiong.

**Formal analysis:** Gang Wang.

**Funding acquisition:** Gang Wang.

**Investigation:** De-fei Xiong, Yi-yang Zhang, Zhi-chao Wang, Gang Wang.

**Methodology:** Yi-yang Zhang, Yuan-ming Zheng, Han-qing Zhang.

**Project administration:** De-fei Xiong, Yi-yang Zhang, Gang Wang.

**Resources:** De-fei Xiong, Yi-yang Zhang, Zhi-chao Wang, Yuan-ming Zheng, Han-qing Zhang.

**Supervision:** Gang Wang.

**Visualization:** De-fei Xiong.

**Writing – original draft:** De-fei Xiong, Gang Wang.

**Writing – review & editing:** Gang Wang.

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
