## [Editor Report · Decision Letter 0]

Dear Dr Wang, 

Thank you for submitting your manuscript entitled "Targeting MED23 enhances the cytosolic viral RNA receptor RIG-I expression and antiviral innate immunity" for consideration as a Research Article by PLOS Biology.

Your manuscript has now been evaluated by the PLOS Biology editorial staff, as well as by an academic editor with relevant expertise, and I am writing to let you know that we would like to send your submission out for external peer review.

Once your full submission is complete, your paper will undergo a series of checks in preparation for peer review. After your manuscript has passed the checks it will be sent out for review. To provide the metadata for your submission, please Login to Editorial Manager (https://www.editorialmanager.com/pbiology) within two working days, i.e. by Nov 15 2024 11:59PM.

Kind regards,

Melissa

Melissa Vazquez Hernandez, Ph.D.

Associate Editor

PLOS Biology

---

## [Decision Letter · Decision Letter 1]

Dear Dr Wang,

Thank you for your patience while your manuscript "Targeting MED23 enhances the cytosolic viral RNA receptor RIG-I expression and antiviral innate immunity" was peer-reviewed at PLOS Biology. It has now been evaluated by the PLOS Biology editors, an Academic Editor with relevant expertise, and by four independent reviewers. I would like to apologize for the extremely long time you waited for the decision. 

In light of the reviews, which you will find at the end of this email, we would like to invite you to revise the work to thoroughly address the reviewers' reports.

As you will see below, the majority of reviewers are positive about the relevance and novelty of the study, yet some concerns were raised during revision. Reviewer 1 would like you to address if MED23 affects FOXO3 mRNA or protein levels, and asks for a more direct evidence of physical interaction between these proteins. Reviewer 2 mentions several mistakes within the article such as not enough detail on the methods, mislabeling or lack of statistical analysis that makes it hard to evaluate the scientific rigor. The reviewer also asks that you perform plaque assays and ELISAs to confirm viral load quantification and cytokine release. Reviewer 3 says the model is oversimplified and might be missing the role of Med23 in regulating type I IFN directly, ans asks that you perform un-biased ChIP-seq. Reviewer 4 questions if the observed effects are due to the control of Med23 in RIG-I transcription, or as mentioned by Reviewer 3, indirectly by mediating IFN. The reviewer gives several suggestion to address this concern.

IMPORTANT: after discussion with the Academic Editor and the reviewers, we agree that the experiments to clarify whether its a direct effect on transcription or an indirect effect by IFN induction should be performed. While we think the other experiments will enhance the mansucript, we will leave to your discretion performing them. 

Given the extent of revision needed, we cannot make a decision about publication until we have seen the revised manuscript and your response to the reviewers' comments. Your revised manuscript is likely to be sent for further evaluation by all or a subset of the reviewers.

**IMPORTANT - SUBMITTING YOUR REVISION**

*Re-submission Checklist*

*Published Peer Review*

*PLOS Data Policy*

*Blot and Gel Data Policy*

Sincerely,

Melissa

Melissa Vazquez Hernandez, Ph.D.

Associate Editor

PLOS Biology

REVIEWERS' COMMENTS

Reviewer #1: 

Using a combination of biochemical and molecular analyses as well as ex vivo cultured mammalian cells and tissue-specific MED23 knockout mouse models, Xiong et al. provide important insights into the role of Med23 and FOXO3 in regulating the innate immune response via RIG-I in mammals. The manuscript is well-written, with a logical experimental design and solid data. The findings are novel and improve our understanding of the molecular mechanisms underlying innate immunity. Below are a few points that could further strengthen the study.

Major Points:

First, it is unclear why MED23 deletion reduces FOXO3 binding to the RIG-I promoter (Fig. 6G). Does the loss of MED23 affect FOXO3 mRNA or protein levels? Additional evidence or discussion to clarify this mechanism would help the understanding of this observation.

Second, the interaction between Med23 and FOXO3 is supported by co-IP experiments. However, co-IP does not exclude the possibility of indirect interactions. Given the importance of this point, additional experiments, such as a GST pulldown assay or alternative methods, could provide more direct evidence for the physical interaction between these two proteins.

Minor Points:

First, the rescue experiments involving ectopic Med23 expression in Med23 KO cells are critical. However, it is unclear whether the ectopically expressed Med23 is comparable in levels to endogenous Med23. This applies to Fig. 1C and Fig. S3H. Clarifying this point would strengthen the conclusions.

Second, the authors conclude that "Med23 deficiency enhances the innate immune response for viral clearance and minimizes tissue damage in vivo" (Fig. 3; page 11). This conclusion might be an overinterpretation, as the observed effects could also result from Med23's effect on viral replication. Additional evidence to distinguish between these possibilities or revised wording would be appropriate.

Third, on page 12, the authors stated, "These downregulated genes in Med23ΔM BMDMs were enriched in the cell cycle and DNA replication, but Med23 deficiency did not affect the population of BMDMs (data not shown)." It is unclear what analyses were conducted to conclude that Med23 mutation has no effect on BMDM populations. Did the authors assess cell cycle profiles to determine the absence of effects on cell cycle progression? If so, further explanation of how these findings align with the observed gene expression changes would be valuable. Adding these data in the Supplementary data would be helpful.

Fourth, the authors noted that "numerous Foxo3 target genes were upregulated in Med23 knockout cells" (page 18). Including a Venn diagram to show the overlap of these genes with FOXO3 targets and their relative proportions would provide a clearer representation of these findings.

Finally, in the Discussion section, the authors could elaborate on the potential role of other Mediator subunits in the described processes. For instance, in Med23 KO cells, does FOXO3 still require interaction with the remaining subunits of the Mediator complex to activate the transcription of RIG-1 gene? Additionally, how does the Mediator complex (with Med23) shift from repressing to activating RIG-1 transcription? The authors could also discuss whether Med23 binding to FOXO3 or the rest of the Mediator complex is regulated by viral infection. Furthermore, it would be helpful to discuss whether the model presented in Fig. 7 also explains the effects of Med23 loss on TBK1 and IRF3 phosphorylation, as well as IRF3 nuclear translocation, as shown in Fig. 1.

Reviewer #2: 

The study focuses on an intriguing aspect of innate immune regulation during RNA virus infection, identifying Med23 as a key regulator of the RIG-I-dependent antiviral response. The authors utilize a range of methodologies, multiple cell lines, and a transgenic mouse model to provide insights into this critical regulatory mechanism. However, the manuscript is hindered by several issues, including insufficient details on experimental rationale and design, statistical analyses, inconsistent labeling, and mislabeled panels. These issues make it significantly difficult to evaluate the study's quality and scientific rigor.

Major Points:

1. Critical experimental designs and conditions, such as VSV infection and poly(I:C) treatment, are not adequately described. These methods are fundamental to stimulating RIG-I and must be clearly stated for scientific rigor. Additionally, statistical analysis needs to be explicitly described in the figure legends, along with the number of experimental repeats. The lack of this information compromises the scientific integrity and hinders a thorough review of the manuscript.

Here are some examples,

Fig. 1C: Reconstitution methodology (e.g., transfection) is not described.

Experimental conditions (MOIs, treatment times) are absent for Figs. 1D-E.

Fig. 5 lacks information on siRNA procedures, CHX concentration, and other experimental details.

Fig. 6A: Foxo3 transfection details (e.g., plasmid amount, time point) and the units in the panel are missing.

Fig. 6F: The rationale behind analyzing additional proteins (Med1, Med16, etc.) is not explained in the text or figure legend.

Fig. 6H: Experimental design details are absent, making interpretation difficult.

No description of the y-axis label "relative mRNA level" in qRT-PCR panels.

2. It is unclear whether stable Med23 KO cells, such as MEFs, exhibit baseline levels of certain IFN-I or cytokines. For instance, in Figs. 1A and 1B, Il6 shows a higher level but no significant increase in response to VSV and poly(I:C) treatments. Clarification is needed.

3. Although qRT-PCR is used to assess viral load in Figs. 1, 2F-G, and 3D, as well as cytokine mRNA levels, plaque assays, and ELISA, are more reliable and should be included to confirm viral load quantification and released cytokines.

4. Figs. 1H and 1J: A strong increase in nuclear IRF3 after VSV infection is observed. However, this change is inconsistent with Fig. 1H. The rationale for this discrepancy needs to be addressed.

5. In Fig. S1J, Med23 is induced by poly(I:C), but this observation is inconsistent with Fig. 1H. This contradiction requires clarification.

6. Fig. 2A: The presence of three lanes per mouse is unexplained. Additionally, KO cells exhibit basal Med23 protein expression. Is this due to incomplete knockout or contamination by non-myeloid cells? This requires explanation.

7. Fig. 1A shows strong increases in IFN-I and cytokine levels in MEF KO cells compared to WT, whereas Figs. 2B-C show moderate increases in BMDM and PEM. If infection conditions are comparable, why are the effects less pronounced in BMDM and PEM? Missing VSV infection details impede proper evaluation.

8. Fig. 2E: VSV infection conditions and infection times are not described. Furthermore, why does VSV infection reduce IRF3 levels in this context? The basal expression of Med23 in KO cells also needs to be addressed.

9. Fig. 3: Given a stronger innate response, does Med23 KO protect mice from VSV-induced diseases, such as weight loss or survival deficits?

10. Figs. S3F-G: If the KO cells exhibit higher basal ISRE activity, the statistical significance should be shown, and its implications discussed.

11. Fig. 5I and 6G: Limited information about the ChIP experiments is provided. There is no description of why these sites were chosen for analysis and the location of the promotor. More importantly, why are the sites analyzed different from these two experiments? 

12. Fig. 5 and Fig. S4: Immunoblotting results show consistently higher basal RIG-I protein levels in Med23 KO cells across experiments, yet qRT-PCR does not reflect increased mRNA levels. This suggests that there are additional regulations that cause higher basal protein levels. Do the authors have any comments on the higher protein levels?

Minor Points:

1. Fig. 1 lack WB for Med23 KO mice.

2. Fig. S1:Clarify whether the KO cell lines are single clonal.

3. Use consistent terminology: Type I IFN or IFN-I.

4. Protein markers for molecular weights are missing in immunoblots.

5. In Fig. 1C, "Gapdh" should be capitalized.

6. The statement in Line 219 regarding elevated IFN-β and Il6 mRNA levels in Med23ΔM mice liver is inconsistent with Fig. 3C, where no statistical significance is shown.

7. Rsad2 is not a cytokine (Line 262).

8. Fig. 4: Gene ontology analysis should be included to provide insights into the role of the downregulated genes in Med23 KO cells, with or without VSV infection.

9. Fig. 4E: Data presented as a heatmap would benefit from statistical analysis and fold change representation for greater interpretability.

10. Fig. 6G is mislabeled as Fig. 6I in the figure legend.

11. Line 280: The statement "Med23 interferes with innate immune signaling by limiting ISG expression during VSV infection" should be revised for clarity to: "Med23 interferes with innate immune signaling, resulting in limited ISG expression during VSV infection."

Reviewer #3: 

The manuscript by Xiong et al. reports the transcriptional level regulation of RNA virus sensor RIG-I by Mediator complex subunit 23 (Med23). Using different cell lines and primary cells from mice, they provided convincing evidence that Med23 negatively regulates type I interferon, inflammatory cytokines and chemokines in the context RNA virus infection or mimicry. In vivo, the authors provided a physiological relevance that Med23 represses type I interferon thus dampening anti-viral response. Overall, they presented a model that Med23 collaborates with Foxo3 to suppress RIG-I leading to downregulated type I interferon response.

Although transcriptional level regulation of innate immune sensors, inflammatory cytokines and chemokines by mediator seems novel, they author oversimplified the model by drawing a single direction arrow from RIG-I to type-I interferon and other ISGs. However, since RIG-I (DDX58) is also an ISG, the reduced level of type I interferon is expected to lead to lower RIG-I level. Biased by their model, the authors did not address the possibility that Med23 directly regulate type I interferon or other inflammatory cytokines. Indeed, in the absence of viral infection, Med23ΔM already showed an increase of various ISGs (Figure 4B and E), indicating the regulation of ISGs might be independent of RIG-I's recognition of viral RNA. Given the broad role of mediator complex, the major scientific question of this manuscript will be better addressed with unbiased ChIP-seq approach. 

Beyond the major defect in the overall model. There are multiple defects that impact the evaluation of this work:

(1) Figure 1A figure caption and actual panel are mismatched.

(2) Not rigorous about statistical test and labeling of the significance. First, they did not mention the correction method used for multiple comparisons. Second, the labeling of the significance is arbitrary. For example, in figure 5c, non-significant is labeled for the upper panel but the label is missing for the lower panel siRIG-I #2. Third, some panels are presented with individual values, the others are not.

(3) Using qPCR as readout for Ifnb1 and interferon alpha, the name of the gene/transcripts are not labeled correctly. This is a big issue for interferon alpha, because there are 14 mouse interferon alpha subtypes. 

(4) Line 146 says "immune escape of VSV" but the description of the viral load data started in the next paragraph.

(5) Line 184 use "macrophage-specific" to describe LysMCre.

(6) Is there any immunopathology in vivo, due to increased interferon and inflammatory response in Med23ΔM with and without viral infection?

(7) Line 262: Cxcl9, Cxcl10 are chemokines and Rsad2 is SAM family enzyme. Not cytokines

(8) Figure 4. It looks strange that the -log10(P) in D is close to 1 (P close to 0.1) Given C looks comparable with A, B and D should be in similar scale.

(9) RIG-I is just one possible sensor. How about other viral sensors like cGAS, TLR3. Does Med23 specifically suppresses one sensor but not the other?

(10) Figure 6F. The right panel, IP: HA, Foxo3 seems to have higher background in WT relative to other mediator or Cdk8 making the statement about the interaction less convincing. 

Reviewer #4: 

Innate immune sensors that induce type I IFNs play important protective roles in infection and cancer but may contribute to deleterious inflammation in other disease settings. As such, regulation of these sensors that include RIG-I is crucial. Tight control of RIG-I expression and activity by posttranslational modifications is well established and has been widely studied. Transcriptional control has received less attention but may be equally important.

By establishing that Med23 controls RIG-I gene transcription, this manuscript provides new and potentially highly relevant insights. Overall, the data are of high quality and support the conclusions well, and the manuscript is well written with nicely laid out figures. In my view, this manuscript is a good candidate for publication in PLoS Biology if the points below can be addressed by major revision.

My key concern relates to whether the observed effects are specifically due to Med23-mediated control of RIG-I transcription. Indirect effect, e.g. via IFN mediated induction of RIG-I transcription, seem possible, as are broader effects of Med23 on other pathways. Points 1-6 below include suggestions to address these issues.

Specific points

1. RIG-I protein expression appears increased in uninfected cells lacking Med23, e.g. Fig 5A (BMDMs), S4A/B (MEFs) or S4C (RAWs). Is this effect reproducible? Is RIG-I mRNA level increased in uninfected cells (e.g. in Fig 4A)?

2. RIG-I is encoded by an ISG. Therefore, the authors should test if RIG-I expression is controlled by Med23 in cells lacking IFNAR, with and without infection.

3. The RIG-I promoter analysis (Fig S5) would benefit from inclusion of the ISREs and other elements. Do these overlap with Foxo3 sites?

4. To demonstrate specificity for RIG-I, the authors should induce IFNs in WT and Med23 KO cells using DNA transfection or with cGAMP, which both trigger the cGAS-STING pathway.

5. Are other innate immune sensors controlled by Med23? Please analyse RNAseq data in Fig 4 for other sensors, including MDA5 and LGP2.

6. Fig 5A. Please include dose-titrations to ensure that the response is not maximal (which may mask differences between WT and KO).

7. Fig 5D. Med23 protein levels appear to be increased in RIG-I-depleted cells. Is this reproducible? If yes, can the authors please comment on this apparent reciprocal regulation? 

8. Fig 5E. The authors may wish to consider measuring RIG-I mRNA decay in WT and KO cells after ActD treatment. If the authors' model is correct, mRNA half-live is expected to be unchanged in KO cells.

9. Fig 5G-H. The RIG-I promoter luciferase reporter is not described in the methods and main text, and no citation is provided. Is this a new reagent generated in this study? If yes, please provide a detailed description and validation data for this reagent. Alternatively, if this is a published reagent, please provide a brief description and relevant citation(s).

10. Figure legends should include information on the number of technical and biological repeat experiments for each panel, and whether pooled or representative data are shown. Bars and error bars need to be defined. Why do some graphs show individual data points (e.g. in Fig 5, panels F, G, H), whereas other graphs do not (e.g. in Fig 5, panels A, C, I)? 

Minor points

11. Many of the reviews cited in the introduction are outdated, incl. refs. 1, 3-6, 10, and 18-19. I recommend replacing with up-to-date reviews from the last few years.

12. Lines 109-111, 113-114, 120-124, 198-199, 201-202, 263-266, 319-321, 338. Please indicate in text that mRNA levels are measured here.

13. Lines 160-161. The Med23 western blot mentioned as panel (A) in the legend is missing from the figure.

14. Line 222 "the fold change in" should be deleted as panel 3D shows relative expression, not fold change.

15. Fig 3E. Can the authors quantify / score pathology in different sections from different animals to demonstrate reproducibility?

16. Fig 4E and lines 294/295. Please clarify whether these data are with or without VSV.

17. Line 273: use "ISGs" instead of "ISG genes"

18. Line 372: there is no panel S5C in Fig S5.

19. Fig 6D. Please include total TBK1 blot, and ideally also p-IRF3 and total IRF3 blots.

20. Line 431. "conditional" (not "conditioned")

---

## [Editor Report · Decision Letter 2]

Dear Dr Wang,

Thank you for your patience while we considered your revised manuscript "Targeting MED23 enhances the cytosolic viral RNA receptor RIG-I expression and antiviral innate immunity" for publication as a Research Article at PLOS Biology. This revised version of your manuscript has been evaluated by the PLOS Biology editors, and the Academic Editor.

Based on our Academic Editor's assessment of your revision, we are likely to accept this manuscript for publication, provided you satisfactorily address the remaining editorial points. Please also make sure to address the following data and other policy-related requests.

a) We routinely suggest changes to titles to ensure maximum accessibility for a broad, non-specialist readership, and to ensure they reflect the contents of the paper. In this case, we would suggest a minor edit to the title, as follows. Please ensure you change both the manuscript file and the online submission system, as they need to match for final acceptance:

"Mediator complex subunit MED23 dampens antiviral innate immunity by restricting RIG-I expression"

b) Please note that per journal policy, the model system/species studied should be clearly stated in the abstract of your manuscript. Currently, it is not clear in your abstract which RNA viruses were tested. 

c) The Ethics statement needs to be a separate, independent (and the first) subheading in the Material & Methods section. It must include the full name of the IACUC/ethics committee that reviewed and approved the animal care and use, as well as the protocol/permit/project license number. Currenlty you are missing the permit number and it is under the subsection "mice".

https://journals.plos.org/plosbiology/s/ethical-publishing-practice

d) Thank you for providing the numerical values and raw data for most of the figures. However, we would also need the numerical values for Figures 4ACE, S3ACD

e) Please cite the location of the data clearly in all relevant main and supplementary Figure legends, e.g. “The data underlying this Figure can be found in S1 Data” or “The data underlying this Figure can be found in https://doi.org/10.5281/zenodo.XXXXX”

f) Currently the GEO you provided is private. Please make sure to make this public. 

g) Thank you for providing the "raw" images of the blots and gels. However, we require the original, uncropped and minimally adjusted images supporting all blot and gel results reported. Currently, the figures provided seem to be from cropped gels. 

Please carefully read our guidelines for how to prepare and upload this data: https://journals.plos.org/plosbiology/s/figures#loc-blot-and-gel-reporting-requirements

h) For figures containing FACS data (Figures 1MN, S1PQ), please provide the FCS files and a picture showing the successive plots and gates that were applied to the FCS files to generate the figure. We ask that you please deposit this data in the FlowRepository (https://flowrepository.org/) and provide the accession number/URL of the deposition in the Data Availability Statement in the online submission form.

i) Please ensure that your Data Statement in the submission system accurately describes where your data can be found and is in final format, as it will be published as written there.

j) Per journal policy, if you have generated any custom code during the course of this investigation, please make it available without restrictions upon publication. Please ensure that the code is sufficiently well documented and reusable, and that your Data Statement in the Editorial Manager submission system accurately describes where your code can be found.

We expect to receive your revised manuscript within two weeks. 

*Published Peer Review History*

*Press*

Sincerely,

Melissa

Melissa Vazquez Hernandez, Ph.D.

Associate Editor

PLOS Biology

---

## [Editor Report · Decision Letter 3]

Dear Dr Wang,

Thank you for the submission of your revised Research Article "Mediator complex subunit MED23 dampens antiviral innate immunity by restricting RIG-I expression" for publication in PLOS Biology. On behalf of my colleagues and the Academic Editor, Frank Kirchhoff, I am pleased to say that we can in principle accept your manuscript for publication, provided you address any remaining formatting and reporting issues. These will be detailed in an email you should receive within 2-3 business days from our colleagues in the journal operations team; no action is required from you until then. Please note that we will not be able to formally accept your manuscript and schedule it for publication until you have completed any requested changes.

Thank you for the clarification regarding the WB images. 

IMPORTANT: While, there is no need to wait for Flow Repository. However, please provide the FCS files, you can upload them to our system in zip format or to a depository like Zenodo. I have asked my colleagues to include this request alongside their own.

PRESS

Sincerely,

Melissa

Melissa Vazquez Hernandez, Ph.D., Ph.D.

Associate Editor

PLOS Biology
